# DeepCOVID-Fuse: A Multi-Modality Deep Learning Model Fusing Chest X-rays and Clinical Variables to Predict COVID-19 Risk Levels

**DOI:** 10.3390/bioengineering10050556

**Published:** 2023-05-05

**Authors:** Yunan Wu, Amil Dravid, Ramsey Michael Wehbe, Aggelos K. Katsaggelos

**Affiliations:** 1Department of Electrical and Computer Engineering, Northwestern University, Evanston, IL 60201, USA; aggk@eecs.northwestern.edu; 2Department of Computer Science, Northwestern University, Evanston, IL 60201, USA; amildravid2023@u.northwestern.edu; 3The Division of Cardiology, Department of Medicine and Bluhm Cardiovascular Institute, Northwestern Memorial Hospital, Chicago, IL 60611, USA; ramsey.wehbe@northwestern.edu

**Keywords:** COVID-19, risk level prediction, multi-modality, fusion CNNs, CXRs, clinical variables

## Abstract

The COVID-19 pandemic has posed unprecedented challenges to global healthcare systems, highlighting the need for accurate and timely risk prediction models that can prioritize patient care and allocate resources effectively. This study presents DeepCOVID-Fuse, a deep learning fusion model that predicts risk levels in patients with confirmed COVID-19 by combining chest radiographs (CXRs) and clinical variables. The study collected initial CXRs, clinical variables, and outcomes (i.e., mortality, intubation, hospital length of stay, Intensive care units (ICU) admission) from February to April 2020, with risk levels determined by the outcomes. The fusion model was trained on 1657 patients (Age: 58.30 ± 17.74; Female: 807) and validated on 428 patients (56.41 ± 17.03; 190) from the local healthcare system and tested on 439 patients (56.51 ± 17.78; 205) from a different holdout hospital. The performance of well-trained fusion models on full or partial modalities was compared using DeLong and McNemar tests. Results show that DeepCOVID-Fuse significantly (*p* < 0.05) outperformed models trained only on CXRs or clinical variables, with an accuracy of 0.658 and an area under the receiver operating characteristic curve (AUC) of 0.842. The fusion model achieves good outcome predictions even when only one of the modalities is used in testing, demonstrating its ability to learn better feature representations across different modalities during training.

## 1. Introduction

Coronavirus disease 2019 (COVID-19) has been heavily straining the healthcare systems of countries across the world, with over 500 million cases and 6 million deaths as of July 2022 [1]. Reverse-transcription polymerase chain reaction (RT-PCR) is the current gold standard for the diagnosis of COVID-19. However, the use of RT-PCR for COVID-19 diagnosis is limited to authorized, trained clinical laboratory personnel and patients with suspected COVID-19, which can create bottlenecks in the testing process. Results can take more than 24 h to produce, leading to delays in patient care and allocation of resources [2]. Previous studies have shown that chest radiographs (X-rays) and computed tomography (CT) images can reveal COVID-19 features [3], which can be combined with clinical judgment to make a COVID-19 diagnosis. Artificial Intelligence (AI) algorithms have shown great promise in detecting these signatures from chest X-rays and CT scans [4,5,6,7], enabling faster and more accurate treatment of suspected patients. The use of CT in this domain is restrictive, particularly due to their high cost and longer processing time, whereas X-ray scanning is more commonly conducted and accessible. Nevertheless, relying solely on chest X-rays may not be sufficient to serve as a diagnostic tool for COVID-19, but they can instead be used to inform the diagnosis and flag at-risk patients who may need further testing [8]. This is particularly important in resource-limited settings, where the use of chest X-rays can aid in resource allocation, triage, and infection control.

Many AI models have been proposed for the purpose of COVID-19 detection [9,10,11,12]. For example, Shaheed et al. developed a computer-aided diagnostic scheme that utilizes extracted features from transformers and a random forest classifier on CXRs for automatic recognition of COVID-19 and pneumonia [13]. However, their use of small and biased public datasets raises skepticism about their deployment [14]. In contrast, DeepCOVID-XR, which proposes an ensemble of convolutional neural networks (CNNs), serves as one of the first works trained and tested on a large clinical dataset for COVID-19 detection [4]. It has been found to be more robust to biases present in models trained on public data. With rapid testing becoming more readily available than at the beginning of the pandemic, more efforts can be made to serve those who are infected. Furthermore, identifying the severity and prognosis of infected individuals can aid in triaging and allocating resources appropriately. However, while many published AI algorithms have focused on better detection of COVID-19, risk stratification of confirmed COVID-19 subjects remains relatively unexplored. To address this gap, we propose DeepCOVID-Fuse, a model that fuses clinical variables from electronic health record (EHR) data and image features from CXRs to categorize infected COVID-19 patients into low-, intermediate-, and high-risk classes. Previous work on fusion and risk prediction has included utilizing the fusion of different image feature representations for COVID detection [15], tackling binary risk prediction with patient characteristic data [16] and clinical features [17], predicting the chance of survival and kidney injury with tabular clinical and biochemistry data [18], and utilizing gene enrichment profiles from blood transcriptome data to stratify COVID-19 patients [19]. Our DeepCOVID-Fuse model ensembles three different architectures with image and tabular clinical data, providing accurate fine-grained risk predictions for COVID-19 patients. Notably, we thoroughly compare the performance of fusion models trained on multiple modalities but tested on one or a subset of modalities, and find that testing the fusion model even with a missing modality still provides more informative predictions than networks trained on a single modality.

## 2. Materials and Methods

### 2.1. Patients

This study is based on a cohort of 2085 COVID-19 patients from over 20 different sites across Northwestern Memorial Health Care System. All patients were tested positive from February 2020 to April 2020 and their corresponding electronic health records (EHR) were collected with a positive reverse transcription polymerase chain reaction (RT-PCR). Unlike a previous study [4], which used CXRs of both COVID-19-negative and positive subjects to build models for COVID-19-positive prediction, our study focuses exclusively on the initial CXR taken after the first inpatient admission of each COVID-19-positive subject. Furthermore, we broaden the scope of our investigation by incorporating clinical variables from subjects’ EHRs to make risk predictions for COVID-19 subjects. Specifically, each of the COVID-positive patients was categorized into low-, intermediate-, or high-risk classes. These three classes correspond, respectively, to (1) hospital length of stay (LoS) of less than one day, (2) hospital LoS greater than one day but no death or admittance to the ICU, and (3) death or admittance to the ICU, as documented in the patients’ EHRs.

### 2.2. CXRs Acquisition and Preprocessing

CXRs images were preprocessed in accordance with metadata using appropriate windowing operations. The grayscale images were first converted to 3-channel RGB images (with identical R, G, and B planes) as this is the typical input of deep learning models. To remove unnecessary background and focus more on lung features, images were then center-cropped using a UNet-based algorithm [20], which was pre-trained on the public CXR dataset [21,22] to segment lung fields. Finally, all cropped images were resized to a resolution of 224 × 224 pixels, scaled to a range of 0 to 1 by dividing each pixel value by 255 (8-bit images), and normalized using ImageNet’s mean and standard deviation before being fed into the model. This preprocessing was applied to all training, validation, and test sets.

### 2.3. Clinical Data Processing

Clinical variables were obtained from each subject’s EHRs across different categories: basic demographic information, laboratory results, comorbidities, electrocardiogram (ECG), and modified early warning score (MEWS). To preprocess the data, we first matched each subject’s first CXR with its temporally closest EHR within 24 h. We then discarded features that were missing more than 40% of their entries. The remaining features were classified into three types for preprocessing, namely, binary, categorical, and continuous, as shown in Appendix A. Specifically, for binary features, such as comorbidities, missing values in the training set were set to non-existent. For multi-class features, such as race and smoking status, missing values were set to an additional unknown class, and all classes were converted to one-hot vectors. For continuous features, missing values were imputed using the mean computed from the training set and all features were scaled to the range of 0 to 1 using min–max normalization. The mean value of each clinical feature on the training set was applied to the validation and test sets for normalization. The details of all selected clinical features are provided in Appendix A.

### 2.4. Model Details

The DeepCOVID-Fuse is a combination of three fusion neural network architectures that were trained using a weighted ensemble approach to accurately classify COVID-19 patients into three risk categories (i.e., low risk, intermediate risk, and high risk), as shown in Figure 1. Each individual network consists of two branches: the CXR image branch and the clinical variable branch. In comparison to our previous model DeepCOVID-XR [4], where six different networks were ensembled—DenseNet-121 [23], ResNet-50 [24], InceptionV3 [25], Inception-ResNetV2 [26], Xception [27], and EfficientNet-B2 [28]—DeepCOVID-Fuse is designed to balance efficiency with accuracy by utilizing only three CNNs for CXR image processing. These three CNNs are chosen as the CXR image branch to process 224 × 224 chest X-rays, namely EfficientNet-B2, ResNet50, and DenseNet-121. To each network, a fully connected layer was added to adjust the feature dimension of the image branch, followed by a dropout layer to prevent overfitting.

Specifically, the clinical variable branch includes a fully connected layer designed to process 99 clinical features from tabular EHR data, followed by a dropout layer. Overall, the two branches were fused together using a concatenation layer, followed by two fully connected layers and a three-class output node with a softmax activation function for final classification. Further information on the hyperparameters can be found in Appendix A. The training process of the entire framework for the two network branches consists of two steps. First, the weights of the image branch were initialized with the corresponding weights from DeepCOVID-XR, while the clinical variable branch was randomly initialized in accordance with TensorFlow standards. During this stage, only the clinical variable branch and the fusion layers were trained, while the convolutional layers of the image branch were frozen. In the second stage, after early stopping concluded the first stage of training, all layers were unfrozen and fine-tuned. Finally, the outputs of each of the three models (i.e., probabilities after softmax) were averaged for the final prediction.

### 2.5. Statistical Analysis

The performance of different models was evaluated by calculating various metrics such as overall accuracy, precision, recall, F1 score, MCC (The Matthews correlation coefficient), and AUC (area under the receiver operating characteristic (ROC) curve). To ensure reliability, each experiment was run independently five times, and 95% confidence intervals were obtained. MCC is a useful metric for multi-class classification as it considers true positives, true negatives, false positives, and false negatives, making it more suitable for imbalanced classes and providing a comprehensive understanding of the model’s performance. McNemar’s test [29] was performed for pairs of models to compare the accuracy, precision, recall, and F1 score, and the DeLong test [30] was performed to compare the AUCs of different models. A *p*-value < 0.05 was considered statistically significant.

## 3. Results

### 3.1. Experimental Design

A total of 2085 subjects were included in this study. The demographic distribution details are shown in Appendix A. A three-class outcome was predicted for each COVID-19 subject, i.e., low risk (L), intermediate risk (I) and high risk (H). The data split follows the approach used in our previous work [4], where the training and validation sets are sourced from multiple institutes, while the test set is obtained from a separate, different institute. Since only the initial CXR after each COVID-19 subject’s first inpatient admission was considered, all experiments had a total of 2085 images, of which 1657 (L: 476, I: 663, H: 518; Mean age: 58.30 years ± 17.74 (standard deviation); Female: 807) were used for training and 428 (L: 119, I: 176, H: 133; 56.41 ± 17.03; 190) for validation; the same cohort applies to clinical features. A separate hold-out test set of 439 subjects (L: 101, I: 193, H: 145; 56.51 ± 17.78; 205) from a different hospital were used to evaluate model performance.

Overall, three types of models were evaluated and compared in this study, including (1) the fusion models trained on CXRs and clinical features with different feature size combinations (feat_dim in Figure 1) from the image and feature branches, (2) the same models trained on CXRs only with the image branch and (3) the models trained on clinical features only with the feature branch. Additionally, for (1), we compared model performances of three fusion models individually, and the ensemble of all three. For (2), to show the effect of the fusion model, we evaluated the fusion model on CXRs as input only. Likewise, for (3), we evaluated the fusion model on clinical features as input only and compared the result with those trained directly on some machine learning algorithms. Further experimental details are included in the Appendix A. All experiments were run independently five times to account for model variability. The models were trained and evaluated using Tensorflow 2.0 in Python 3.6 on a single GPU (NVIDIA TITAN V).

### 3.2. Performance of DeepCOVID-Fuse

The performance of each individual fusion model and an ensemble of all models on the testing set are compared in Table 1. Overall, the ensemble model significantly outperformed all individual models on this COVID-19 risk prediction task, achieving an accuracy of 0.658, a recall of 0.660, a precision of 0.689, an F1 of 0.660, an MCC of 0.640, and an AUC score of 0.842. Notably, for all individual models, ablation studies of different feature dimensions from CXRs and clinical variables showed that models with higher proportional features from the clinical branch than the CXR branch (i.e., CXRs: clinical = 64:128) achieved better model predictions than equal (i.e., 128:128) or lower fractions (i.e., 1408:128). Furthermore, the fusion model with a DenseNet architecture had the best performance of AUC from 0.814 to 0.824, followed by a ResNet architecture from 0.794 to 0.815 and an EfficientNet architecture from 0.794 to 0.805. In addition, we analyzed the performance of the models across different age groups by categorizing the subjects into four groups: ages 20–40, 40–60, 60–80, and 80–100, as presented in Appendix A. The findings suggest that our proposed model performs well across different age groups. However, the results indicate that the model performs optimally in the middle age group, followed by the younger and older age groups.

### 3.3. Comparison of Image-Only with Fusion-Image-Only

To show the importance of fusion, the performance comparison between the model trained and tested on CXR images only (Image-only) and the model trained on the fusion model (i.e., both CXRs and clinical variables) but tested on CXR images only (Fusion-image-only) are provided in Table 2. Combined with the results in Table 1, the fusion model with additional clinical variables significantly improved COVID-19 risk prediction compared to the Image-only model. Notably, even without the clinical variables, the well-trained fusion model outperformed the Image-only model on the same CXR-only test set. Specifically, for three individual models, the well-trained fusion model improved the accuracy by 0.008~0.011, the recall by 0.004~0.012, the precision by 0.008~0.043, the F1 by 0.007~0.013, the MCC by 0.009~0.020, and the AUC by 0.009~0.016. Additionally, heatmaps generated from the Image-only and Fusion-image-only models using gradient class activation maps (Grad-CAM) are provided in Figure 2 to visualize the salient features of each CXR used by the model for COVID-19 risk level classification. For correct risk-level predictions, these heatmaps highlight abnormalities in the lungs and demonstrate that the fusion model captures more relevant features for classification than the image-only model. In some cases, where the fusion model made the correct classification and the image-only model misclassified, the heatmaps showed different feature patterns, with the former highlighting lung abnormalities and the latter not.

Typically, the clinical features are partially present. Appendix A illustrates the results of the well-trained fusion model on CXR images with the proportionally increasing clinical variables. The results showed that, as more clinical features are integrated into the model, its performance of the well-trained fusion model on the test set improves. For example, when 80% of the clinical variables are present, and only 20% are missing at random, the fusion model (e.g., with the DenseNet architecture) achieved an accuracy of 0.645, a recall of 0.647, a precision of 0.656, an F1 of 0.644, an MCC of 0.625, and an AUC of 0.816.

### 3.4. Comparison of Feature-Only with Fusion-Feature-Only

We performed the same analysis as described above, by comparing the performance of models trained and tested on clinical features only (Feature-only) with models first well-trained on the fusion model but tested on the clinical features (Fusion-Feature-only). As shown in Table 3, even without CXRs as input, the well-trained fusion model significantly outperformed the Feature-only model with an AUC of 0.733. In addition, we compared the neural-network-based models with several machine learning algorithms trained on the same clinical features, including random forests (RM), quadratic discriminant analysis (QDA), and Linear Ridge (LR) classification. Interestingly, RF achieved the best performance, followed by Fusion-Feature-only and LR, while Feature-only had the lowest performance among all metrics. Furthermore, it is still worth noting that the model in Table 1 combining CXRs and clinical variables still outperformed all results in Table 3.

## 4. Discussion

In this study, we proposed DeepCOVID-Fuse, a fusion model that incorporates clinical variables with CXRs to predict future risks of clinically meaningful outcomes in patients diagnosed with confirmed COVID-19. The fusion model was trained and tested using only the first inpatient admission data of each subject, which has great clinical implications for improving our healthcare management system, particularly in intensive care units. DeepCOVID-Fuse achieved an overall accuracy of 0.658 and an AUC of 0.842 on a hold-out testing set from a separate hospital. We further compared this model with models trained on CXR images only or clinical variables only, and evaluated the performance of DeepCOVID-Fuse when only CXR images or clinical variables were available. To the best of our knowledge, our study is the first to demonstrate the effectiveness of a fusion model, which is well-trained on multiple modalities but is capable of achieving a better prediction performance and generating meaningful visual heatmaps when only one or parts of the modalities were available on CXRs and clinical features.

The aim of our work is to assist with resource allocation by addressing a three-class prediction problem, where the level of risk for COVID-19 patients is determined based on their mortality status, need for mechanical ventilation, ICU admission, and hospital length of stay (LoS). The three classes are categorized as low, intermediate, and high risk. As the demand for hospital capacity is reported to be dramatically increasing during the COVID-19 pandemic [31], predicting ventilator usage or ICU admissions in advance will reduce pressure on hospitalization management. In addition, LoS is critical to the allocation of bed capacity, so we chose a 1-day LoS as the separation to differentiate low and intermediate risk, as only patients with a LoS of more than 1 day needed to be allocated a bed. Furthermore, the results in Table 1 and Table 2 show that the fusion model with the addition of clinical variables significantly improved risk performances over the model trained only on CXRs, indicating that clinical variables are strongly associated with COVID-19 severity. Meanwhile, the performance of the ensemble fusion models being higher than that of each model individually is consistent with the previous study that showed the ensemble model reduces the generalization error of predictions [4].

In most real-world scenarios, it is common for a modality to be missing or incomplete. As such, the fusion model is not guaranteed to utilize inputs from all modalities, i.e., some COVID-19 patients have either CXRs or a subset of clinical data. One study showed that this can be a limitation of fusion models, as predictions can be overly influenced by the most feature-rich modalities leading to poor generalization [32]. However, our study shows that even if only one or partial modalities are available, well-trained fusion models can still achieve better performances than models trained on that single or partial modality alone, as shown in Table 2 and Table 3. Learning correlations across different modalities is the possible explanation for this improved performance. Specifically, since different modalities of a fusion model are simultaneously back-propagated through the loss, they complement each other, so the fusion model is able to learn better latent space representations for each model branch. Therefore, even if only a subset of CXRs or clinical variables are available, fusion models can still play an important role in learning more discriminative features. The experiments in Appendix A further show that once the fusion model was well trained, the model performance continued to improve as long as more clinical variables were available in the test set. This can have significant implications for future medical research, as it gives a strong support to the scenario that if images are provided with more usable information during training, i.e., simple features such as age and gender, even if only images are available at testing stage, a better classification prediction can be achieved compared with that using only images to train and test models.

Heatmaps generated by Grad-CAM provide another perspective on the superiority of fusion models in learning feature representations of CXR images compared to image-only models. As shown in Figure 2e–h, when only CXRs were available, the image-only model misclassified a high-risk subject as intermediate, while the well-trained fusion model made the correct prediction. This can be observed from their respective heatmaps, where the fusion model highlighted discriminative features of the lung, while the other located the wrong area. When making the correct predictions, all heatmaps looked at areas close to the lung, as shown in Figure 2a–d,f–h.

Although previous studies have existed to predict the severity of confirmed COVID-19 patients, our work has a different focus and is unique in many ways. For example, Liang et al. developed a DL-based survival model on a 1D clinical dataset collected at admission to predict the risk of COVID-19 patients being critically ill within 30 days, achieving an overall AUC of above 0.85 [33]. However, they were limited by a lack of clinical datasets, and no imaging data were available. Shamout et al. later proposed a deep learning model using CXRs and routine clinical variables to predict the deterioration risk (i.e., intubation, ICU admission, or mortality) in COVID-19 subjects within 96 h with an AUC of 0.786 [34]. Similarly, Jiao et al. used a DL network combining CXR and clinical data to predict binary outcomes of COVID-19 patient severity (i.e., severe or not), and obtained AUCs ranging from 0.731 to 0.792 [8]. Although two modalities were provided, both studies adopted a late fusion strategy with two independently trained models. In contrast, we trained an end-to-end fusion model that could learn and transfer information between two modalities. A study similar to our work that combined initial CXRs and clinical variables into an end-to-end fusion model to predict mortality in COVID-19 subjects achieved an AUC of 0.82 [35]. However, their model was only trained on 499 subjects with an age range of 21 to 50 years, which may lead to poor model generalization, whereas our model included 2085 subjects of all ages. Another study from Soda et al. developed a multi-branch deep learning framework that combined CXRs and clinical information to predict the clinical outcome (binary: mild or severe) of COVID-19 patients [36]. The model achieved an accuracy of 0.748; however, the study focused solely on the binary outcome with a relatively small sample size of 820 subjects. In addition, Deb et al. proposed a CovSeverity-Net, which uses CXRs to estimate the severity (mild, moderate, severe) of COVID-19 patients [37]. However, unlike our approach, which aims to better assist with resource allocation, Deb et al. included images from all time points, rather than solely using the first inpatient admission of each subject. Most importantly, the focus of this paper is to comprehensively evaluate the statistical and visual performance of fusion models trained on multiple modalities but tested on one or a subset of modalities.

There are some limitations to this study that need to be acknowledged. First, several clinical data in the training dataset are still missing or incomplete. Although we have shown that that not all clinical data is needed in the test set, having a more complete training dataset guarantees a better and more robust model. Second, we did not compare the performance of our fusion model with radiologists, because risk prediction by experts on both CXR and clinical data is challenging and subjective. There is no true, universal ground truth. Next, as shown in Table 3, we found that basic machine learning algorithms, such as random forests, outperformed deep learning-based models, indicating that our fusion model has not yet perfectly extracted features from 1D clinical data. Therefore, future work will explore integrating random forests with deep neural networks to further improve model performance. Lastly, it is worth noting that the current model cannot identify the most relevant features that contribute to patient outcomes, which is important information for clinicians. To address this limitation, we plan to explore new models, such as the merging of random forests with CNNs or incorporating attention mechanisms [38,39], as they may help to predict the importance score of each clinical variable. This information can be valuable for clinicians in understanding the relative importance of different variables in determining patient outcomes.

## 5. Conclusions

In conclusion, we proposed DeepCOVID-Fuse, a fusion model to predict risk levels in COVID-19 subjects using CXRs and clinical variables obtained at their initial inpatient admission. We showed that models combining both CXRs and clinical features outperformed models with only CXRs or clinical variables. Furthermore, we demonstrated that the well-trained fusion model was able to achieve good model performance when only single or partial modality was available. We believe that this work demonstrates that it is possible to predict high-risk patients at admission to further benefit hospital triage systems, and also has the potential to promote the use of fusion models in other fields of medical research. Finally, we have made our codes and model weights publicly available to facilitate future research and enable easy comparison of our model’s performance with others.

## Figures and Tables

**Figure 1 bioengineering-10-00556-f001:**
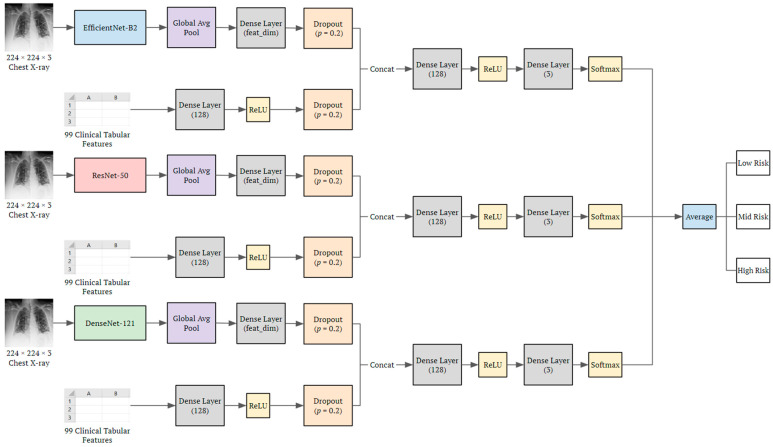
The architecture of the DeepCOVID-Fuse ensemble model. The preprocessed image was fed into three different CNN architectures, followed by a fully connected layer to transform the image feature dimension. The clinical tabular features were fed into a fully connected layer. The features were fused and fed into another fully connected layer, followed by the last classification layer with softmax as the activation function. Feat_dim changes to compare different combinations of features from image and clinical feature branches.

**Figure 2 bioengineering-10-00556-f002:**
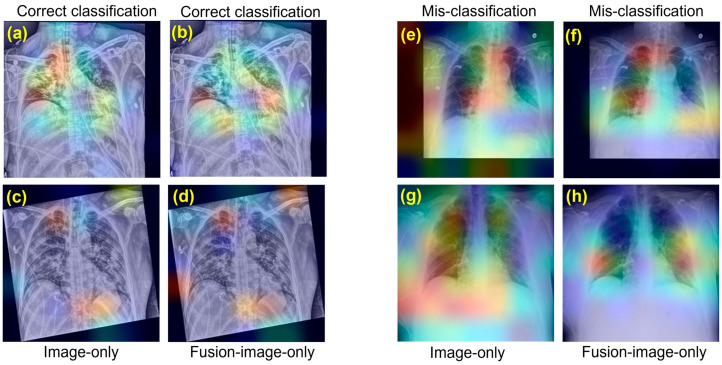
Heatmaps of Gradient class activation maps (Grad-CAM) generated from the model showing the location of important features for high-risk predictions in confirmed COVID-19 patients. The redder the intensity of the heatmap, the more important the feature areas. Heatmaps generated from Image-only models (**a**,**c**,**e**,**g**) and Fusion-image-only models (**b**,**d**,**f**,**h**) are compared for cases of correct (**a**–**d**,**f**,**h**) and incorrect predictions (**e**,**g**). For the same subject using only CXR as model input, the Fusion model made correct predictions, with heatmaps (**f**,**h**) highlighting abnormalities in the lungs, while the Image-only model misclassified the predictions, (**e**,**g**) highlighting unnecessary background.

**Table 1 bioengineering-10-00556-t001:** Performance of fusion models for risk predictions in confirmed COVID-19 subjects on external test sets with different combinations of latent feature sizes from X-rays and clinical variables.

	EfficientNet			ResNet			DenseNet			Ensemble
Latent feature (X_ray × clinical data)	64 × 128	128 × 128	1408 × 128	64 × 128	128 × 128	2048 × 128	64 × 128	128 × 128	1408 × 128	64 × 128
Accuracy	0.618 [0.600, 0.637]	0.622 [0.606, 0.638]	0.626 [0.590, 0.662]	0.628 [0.610, 0.645]	0.630 [0.620, 0.642]	0.611 [0.589, 0.632]	0.658 [0.650, 0.667]	0.638 [0.622, 0.654]	0.640 [0.632, 0.647]	0.658 *
Recall	0.619 [0.600, 0.639]	0.622 [0.606, 0.638]	0.626 [0.590, 0.662]	0.626 [0.595, 0.656]	0.633 [0.623, 0.642]	0.611 [0.589, 0.632]	0.657 [0.649, 0.666]	0.638 [0.621, 0.655]	0.640 [0.632, 0.647]	0.660 *
Precision	0.649 [0.631, 0.666]	0.648 [0.620, 0.676]	0.675 [0.648, 0.702]	0.665 [0.652, 0.678]	0.675 [0.664, 0.685]	0.652 [0.619, 0.685]	0.671 [0.658, 0.684]	0.641 [0.623, 0.659]	0.647 [0.635, 0.659]	0.689 *
F1	0.616 [0.599, 0.633]	0.619 [0.603, 0.637]	0.623 [0.583, 0.663]	0.626 [0.608, 0.645]	0.627 [0.612, 0.642]	0.607 [0.586, 0.627],	0.658 [0.650, 0.666]	0.638 [0.621, 0.655]	0.639 [0.632, 0.647]	0.660 *
MCC	0.607 [0.603, 0.611]	0.614 [0.606, 0.622]	0.617 [0.609, 0.625]	0.618 [0.612, 0.624]	0.620 [0.615, 0.625]	0.601 [0.594, 0.608]	0.635 [0.629, 0.641]	0.624 [0.619, 0.629]	0.626 [0.620, 0.632]	0.640 *
AUC	0.805 [0.798, 0.812]	0.794 [0.778, 0.811]	0.804 [0.780, 0.827]	0.815 [0.804, 0.826]	0.815 [0.809, 0.820]	0.794 [0.782, 0.807]	0.824 [0.822, 0.826]	0.814 [0.797, 0.831]	0.820 [0.805, 0.836]	0.842 *

Notes: Data in parentheses are 95% CIs from five repeated experimental runs. AUC = area under the receiver operating characteristic curve. Latent feature = Image and clinical feature dimensions when concatenated in a fusion model. * *p* value < 0.05 denotes the comparisons are statistically significant.

**Table 2 bioengineering-10-00556-t002:** Performance of models for risk predictions in confirmed COVID-19 subjects on external test sets using only CXRs as model input.

COVID-Level	EfficientNet Image-Only	ResNet Image-Only	Densenet Image-Only	Image-Only Ensemble	EfficientNet Fusion-Image-Only	ResNet Fusion-Image-Only	DenseNet Fusion-Image-Only	Fusion-Image-Only Ensemble
Accuracy	0.582 [0.572, 0.591]	0.614 [0.604, 0.624]	0.615 [0.608, 0.622]	0.621 *	0.593 [0.581, 0.606]	0.625 [0.615, 0.634]	0.623 [0.604, 0.641]	0.632 *
Recall	0.581 [0.572, 0.591]	0.616 [0.604, 0.624]	0.616 [0.607, 0.624]	0.619 *	0.593 [0.582, 0.606]	0.625 [0.615, 0.634]	0.620 [0.608, 0.632]	0.629 *
Precision	0.604 [0.594, 0.614]	0.664 [0.645, 0.683]	0.631 [0.627, 0.634]	0.665 *	0.657 [0.646, 0.667]	0.662 [0.643, 0.681]	0.639 [0.623, 0.647]	0.664 *
F1	0.576 [0.567, 0.586]	0.609 [0.595, 0.624]	0.614 [0.606, 0.621]	0.620 *	0.583 [0.567, 0.600]	0.619 [0.607, 0.631]	0.627 [0.611, 0.639]	0.634 *
MCC	0.553 [0.540, 0.566]	0.587 [0.580, 0.594]	0.602 [0.590, 0.614]	0.608	0.562 [0.553, 0.571]	0.607 [0.594, 0.620]	0.613 [0.605, 0.621]	0.618
AUC	0.769 [0.764, 0.774]	0.798 [0.788, 0.808]	0.781 [0.72, 0.792]	0.807 *	0.781 [0.768, 0.796]	0.807 [0.803, 0.811]	0.797 [0.784, 0.806]	0.813 *

Notes: Data in parentheses are 95% CIs from five repeated experimental runs. AUC = area under the receiver operating characteristic curve; Image-only = models trained and tested with CXRs only; Fusion-image-only = well-trained fusion models but tested with CXRs only. * *p* value < 0.05 denotes the comparisons are statistically significant.

**Table 3 bioengineering-10-00556-t003:** Performance of models for risk predictions in confirmed COVID-19 subjects on external test sets using only clinical variables as model input.

COVID-Level	DNN Feature-Only	Fusion Feature-Only	Random Forests	QDA	Linear Ridge
Accuracy	0.440 [0.432, 0.448]	0.539 [0.525, 0.553]	0.560 [0.553, 0.567]	0.526 [0.519, 0.533]	0.536 [0.527, 0.546]
Recall	0.441 [0.430, 0.449]	0.540 [0.526, 0.555]	0.563 [0.554, 0.569]	0.528 [0.517, 0.539]	0.533 [0.525, 0.541]
Precision	0.193 [0.183, 0.214]	0.567 [0.553, 0.582]	0.588 [0.517, 0.671]	0.532 [0.526, 0.538]	0.544 [0.532, 0.556]
F1	0.269 [0.253, 0.280]	0.560 [0.542, 0.577]	0.573 [0.568, 0.581]	0.479 [0.461, 0.496]	0.536 [0.527, 0.545]
MCC	0.243 [0.230, 0.256]	0.541 [0.529, 0.553]	0.562 [0.550, 0.574]	0.435 [0.421, 0.449]	0.507 [0.497, 0.517]
AUC	0.502 [0.481, 0.522]	0.733 [0.730, 0.737]	0.768 [0.759, 0.777]	0.600 [0.587, 0.613]	0.625 [0.613, 0.636]

Notes: Data in parentheses are 95% CIs from five repeated experimental runs. AUC = area under the receiver operating characteristic curve; QDA = Quadratic Discriminant Analysis; Feature-only = models trained and tested with clinical variables only; Fusion-Feature-only = well-trained fusion models but tested with clinical variables only.

## Data Availability

The code and model weights used in the study are publicly available from the GitHub repository (https://github.com/YunanWu2168/DeepCOVID-Fuse), accessed on 1 May 2023.

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
