# Peer review of "DeepCOVID-Fuse: A Multi-Modality Deep Learning Model Fusing Chest X-rays and Clinical Variables to Predict COVID-19 Risk Levels"

_bioengineering, 2023, doi:10.3390/bioengineering10050556_

Round 1

Reviewer 1 Report

This manuscript proposed a fusion model, DeepCOVID-Fuse, by fusing chest images and clinical features, to predict COVID-19 risks for confirmed subjects. It was shown that this fusing model outperformed any individual model using CXRs or clinical features. The proposed model worked well even if partial modalities were available. The manuscripts contributions are clear. The results achieved are promising. The article was well organized and written. I suggest it be accepted after minor changes.

Deep leaning (DL) models have now been extensively used for prediction of COVID-19 cases in diagnose or subtyping. Although the results have been compared with the results achieved by some general DL models such a Efficient-Net, ResNet and DenseNet, they have not been compared with those models specially aiming at prediction of COVID-19. Most recently, in literature, many novel DL models, including attention-mechanism based ones, have been utilized in COVID-19 related automatic detection and diagnose. The Author is suggested to add this comparison.

Author Response

Reviewer 1:

The authors would appreciate all reviewers for their valuable comments and constructive feedback. We have further modified our work based on their suggestions. We believe we have addressed all the comments made by the reviewers. Detailed responses to the reviewers as well as the list of changes that have been made in the manuscript are provided below.

R1.1:  Deep leaning (DL) models have now been extensively used for prediction of COVID-19 cases in diagnose or subtyping. Although the results have been compared with the results achieved by some general DL models such an Efficient-Net, ResNet and DenseNet, they have not been compared with those models specially aiming at prediction of COVID-19. Most recently, in literature, many novel DL models, including attention-mechanism based ones, have been utilized in COVID-19 related automatic detection and diagnosis. The Author is suggested to add this comparison.

Reply: Thank you for your comment. We appreciate your suggestion to compare our proposed model with recently proposed deep learning models that utilize attention mechanisms for COVID-19 related automatic detection and diagnosis. While we agree that attention-based models have shown promising results in COVID-19 diagnosis, we believe that comparing our proposed fusion model with those models may not be directly relevant. The main objective of this study is to predict the risk level of COVID-19 subjects using CXRs and clinical features and to prove the good performance of the fusion model even with a missing modality during testing. However, we acknowledge the importance of attention-mechanism based models in finding the most relevant features for outcome prediction, so we have carefully planned to include it in our future work.

Changes in the text: We added a paragraph for limitations and future directions in the discussion section based on all reviewers' comments (see Page 10) with new references related to attention mechanisms (see Refs 39, 40).

Reviewer 2 Report

The results are correct and interesting. However, to improve the quality and presentation of the paper, the authors are suggested to address the following comments.

1. The English of the paper should be polished further.

2. Conclusion section needed, Don't combine it with the Discussion section

3.  What are the limitations or future direction of the proposed study?

4) The information on references should be checked double. And references should be in journal format. 

5. Include some more references from 2023, if possible compare 

Minor editing of English language required

Author Response

Reviewer 2:

The authors would appreciate all reviewers for their valuable comments and constructive feedback. We have further modified our work based on their suggestions. We believe we have addressed all the comments made by the reviewers. Detailed responses to the reviewers as well as the list of changes that have been made in the manuscript are provided below.

R2.1: The English of the paper should be polished further.

Reply: Thank you for the suggestion. We have taken steps to address this concern by carefully reviewing the manuscript and making revisions to improve the overall clarity and readability of the text.

Changes in the text: We have meticulously refined the full text of the paper (as shown with track changes).

R2.2: Conclusion section needed, Don't combine it with the Discussion section

Reply: We appreciate your suggestion that we include a separate Conclusion section, and we agree that this would help to clarify our findings and their significance. We revised our manuscript to include a distinct Conclusion section, separate from the Discussion section.

Changes in the text: We separated the Conclusion section from the Discussion section (see Page 10).

R2.3: What are the limitations or future direction of the proposed study?

Reply: Thank you for your suggestion. In terms of limitations, we have included a paragraph discussing these issues. First, several clinical data are still missing or are incomplete in the training dataset. Although we have shown that it is not necessary to have all clinical data in the test set, a more complete training dataset guarantees a better and more robust model. Second, we did not compare the performance of our fusion model with radiologists because risk prediction by experts on both CXR and clinical data is challenging and subjective. There is no true, universal ground truth. Finally, as shown in Table 3, we found that basic machine learning algorithms, such as random forests, achieved a better performance than deep learning-based models, indicating that our fusion model has not yet perfectly extracted features from 1D clinical data. In terms of future directions, we have added some future work in the last paragraph to provide a foundation for further research.

Changes in the text: We added a new paragraph discussing limitations and future work based on all reviewer’s comments (see Page 10).

R2.4: The information on references should be checked double. And references should be in journal format. 

Reply: Thank you for the suggestion. The reviewer is correct. We have carefully reviewed the references and made corrections to ensure that all the information is accurate and up to date. We also made sure to follow the journal's guidelines for reference formatting and have updated the references to meet the journal format requirements.

Changes in the text: We updated the references in the paper to follow the guidelines provided by the journal, and also included new references to further support our findings.

R2.5: Include some more references from 2023, if possible compare. 

Reply: Thank you for your review of our manuscript. We have conducted a thorough search for relevant literature, including publications from 2023, and have identified several additional references that we believe strengthen our discussion and support our findings. We have incorporated these references into the manuscript, as well as provided a comparison with our study, as suggested.

Changes in the text: We incorporated five of the most recent papers published in 2023 that are relevant to this study and compared them in the Discussion section (Page 11-12, Refs [13][17][19][38][40]).

Reviewer 3 Report

In this manuscript, the authors have proposed a model to predict COVID outcomes based on CXR images and clinical data. The subject area is important. I have the following questions for the authors.

1. Predicted model scores are not impressive as similar approaches were taken previously to predict COVID outcomes using clinical data and CXR images and obtained better Accuracy (0.748 ± 0.008) using the same end-to-end approach. (Ref. Soda P et al., AIforCOVID: Predicting the clinical outcomes in patients with COVID-19 applying AI to chest-X-rays. An Italian multicentre study. Medical Image Analysis, 2021.) Authors should compare this study and justify the benefits of their proposed model.

2. Authors should use n-fold (at-least 5-fold) cross validation to test model performance.

3. Alongside other metrics, I recommend authors calculate MCC scores as well.

4. Authors should explain the model performance by segregating the subjects into different age groups to show comparative model performance and justify that the patient age group does not affect model performance.

5. This manuscript would be more interesting if the authors could predict important clinical features that direct better predictability of the model.

6. Authors should provide data and code for reproducibility.

Author Response

Reviewer 3:

The authors would appreciate all reviewers for their valuable comments and constructive feedback. We have further modified our work based on their suggestions. We believe we have addressed all the comments made by the reviewers. Detailed responses to the reviewers as well as the list of changes that have been made in the manuscript are provided below.

R3.1: Predicted model scores are not impressive as similar approaches were taken previously to predict COVID outcomes using clinical data and CXR images and obtained better Accuracy (0.748 ± 0.008) using the same end-to-end approach. (Ref. Soda P et al., AIforCOVID: Predicting the clinical outcomes in patients with COVID-19 applying AI to chest-X-rays. An Italian multicentre study. Medical Image Analysis, 2021.) Authors should compare this study and justify the benefits of their proposed model.

Reply: Thank you for the valuable comments. We acknowledge the study by Soda et al. (2021) and agree that their approach achieved good accuracy in predicting COVID-19 outcomes. However, we would like to emphasize that our proposed model differs in some key aspects that make it a valuable contribution to the field.

One of the main objectives of our paper is to prove the good performance of the fusion model even with a missing modality during testing, which is unique to our study compared to other studies. This is an important consideration, as it reflects the practical limitations that may occur in real-world scenarios where certain modalities may not be available.

Furthermore, our study focuses on predicting the risk level of COVID-19 subjects, which is defined as low, intermediate, and high, whereas Soda et al. had only two groups to predict, i.e., mild and severe. Predicting three classes is inherently more challenging than two, and thus, achieving good accuracy and AUC score is still considered an achievement in our paper. In addition, our objective is to aid in resource allocation by identifying patients who are likely to require more intensive care and resources. To this end, we added a low-level class that includes patients with hospital length of stay (LOS) less than one day, as they do not necessarily require allocation of a bed. As we have discussed in our Discussion section, this new low-level class has practical implications for resource allocation in hospitals, as it allows for better optimization of resources and reduces the burden on the healthcare system. Therefore, we believe that our proposed model has important clinical implications and adds to the existing literature on COVID-19 risk prediction.

The reviewer is correct that it is important to compare our study with the state of the art in the field. We have revised the manuscript to include a comparison between our study and the Soda et al. (2021) study, along with a discussion of the unique contributions of our proposed model.

Changes in the text: We compared our proposed model to the referenced paper (Ref [37]) and discussed our advantages in the Discussion section (see Page 9).

R3.2: Authors should use n-fold (at-least 5-fold) cross validation to test model performance.

Reply: Thank you so much for the comments. We understand that n-fold cross validation is a widely used technique for model evaluation, but we would like to explain that we followed the same data splitting strategy as our previous work [4], which involved combining images from multiple sites, except for images from a single community hospital that were held out as a test set that the algorithm was not exposed to during training or validation. Evaluating our model on this separate test set from another hospital provides a more realistic measure of its generalization ability. In addition, we performed five independent runs to validate our model's performance. The small standard deviation values demonstrate that the model is robust. We believe that this approach provides a rigorous evaluation of our proposed model.

The reviewer is correct that we need to add more details to our data splitting description. We have updated the manuscript to clarify our approach and provide more details on our evaluation strategy.

[4] R. M. Wehbe et al., “DeepCOVID-XR: An Artificial Intelligence Algorithm to Detect COVID-19 on Chest Radiographs Trained and Tested on a Large U.S. Clinical Data Set,” Radiology, vol. 299, no. 1, pp. E167–E176, Apr. 2021, doi: 10.1148/radiol.2020203511.

Changes in the text: We provided further clarification on the reason for our data splitting approach, and we added more details on the experimental setup (see Page 5: experimental design and Page 4: statistical analysis).

R3.3: Alongside other metrics, I recommend authors calculate MCC scores as well.

Reply: Thank you for your comment regarding the use of Matthews correlation coefficient (MCC) as an additional metric for evaluating our proposed model. We agree that MCC is a useful measure of classification performance, especially for imbalanced datasets. Therefore, we have included MCC scores as an additional metric in our revised manuscript, and we have updated the results and discussion section to reflect this addition.

Changes in the text: We added MCC scores for Table 1, Table 2, Table 3, Supplementary Table S1 and Supplementary Table S2.

R3.4: Authors should explain the model performance by segregating the subjects into different age groups to show comparative model performance and justify that the patient age group does not affect model performance.

Reply: Thank you for your valuable comment. We agree that it is important to evaluate the model performance on different age groups and to show that the model is not biased towards a particular age group. In our study, we have considered the age of the patients as one of the important clinical variables in predicting the COVID-19 risk level as we know the older people has high probability of becoming severe, so the age of patients is possible to affect model performance. As the reviewer suggested, we did not explicitly evaluate the performance of the model on different age groups. In order to see if age affects model performance, we segregated the patients into four age groups and evaluated the performance of the model on each group separately on the test set. Our results show that our proposed model performs well across different age groups, although the model performs optimally in the middle age group, followed by the younger and older age groups, respectively.

Changes in the text: We grouped the subjects into four distinct age categories and analyzed their performance in Supplementary Table S2. Additionally, we included the corresponding results in the Results section of the paper (see Page 5).

R3.5: This manuscript would be more interesting if the authors could predict important clinical features that direct better predictability of the model.

Reply: Thank you so much for the comment. Regarding your comment on identifying the most important features of our fusion model, we completely agree that it would add more value and interest to our research. In Table # of our article, we have compared the performance of random forests and CNNs, and found that random forests achieve better results. Moreover, random forests have the ability to output an important score for each feature, which helps to reveal the underlying relationships between features and outcomes. In contrast, CNNs extract features in a "black box" and do not provide these visual interpretations. Therefore, we plan to combine the random forest and CNNs models in our future work, to take advantage of the strengths of both models.

Furthermore, as one of the reviewers pointed out, newer networks such as transformers or attention modules can indicate the importance of features with self-learned attention weights. We have carefully considered this suggestion and we plan to include it in our future work as well.

Changes in the text: We added a paragraph for limitations and future directions in the discussion section based on all reviewers' comments (see Page 10) with new references cited in Refs [39, 40].

R3.6: Authors should provide data and code for reproducibility.

Reply: Thank you for the suggestion. We agreed that open source encourages further research innovation and collaboration. Unfortunately, our dataset is private, and we are still in the process of obtaining permission from our department to make it available to the public. We understand that sharing data is an essential part of scientific research, and we are committed to making it available as soon as possible. Nevertheless, we have made our code and model weights publicly available to facilitate future research and enable easy comparison of our model's performance with others.

The model weights can be downloaded from the shared Drive: https://drive.google.com/drive/folders/1Fq4mO5Ab1ZU8LDyrvy_Qk4USrume0u6B?usp=sharing

The code used in the study is publicly available from the GitHub repository: https://github.com/YunanWu2168/DeepCOVID-Fuse

Changes in the text: We included a new Code Availability section and shared the link to download the model weights and codes.

Round 2

Reviewer 3 Report

I am satisfied with the author's reply.